# Molecular Mechanisms of Phenylpropane-Synthesis-Related Genes Regulating the Shoot Blight Resistance of *Bambusa pervariabilis* × *Dendrocalamopsis grandis*

**DOI:** 10.3390/ijms23126760

**Published:** 2022-06-17

**Authors:** Fengying Luo, Peng Yan, Liling Xie, Shuying Li, Tianhui Zhu, Shan Han, Tiantian Lin, Shujiang Li

**Affiliations:** 1College of Forestry, Sichuan Agricultural University, Chengdu 611130, China; 2019204009@stu.sicau.edu.cn (F.L.); 2020204015@stu.sicau.edu.cn (P.Y.); 201905912@stu.sicau.edu.cn (L.X.); 72111@sicau.edu.cn (S.L.); 10627@sicau.edu.cn (T.Z.); 13722@sicau.edu.cn (S.H.); tlin@sicau.edu.cn (T.L.); 2National Forestry and Grassland Administration Key Laboratory of Forest Resources Conservation and Ecological Safety on the Upper Reaches of the Yangtze River, Chengdu 611130, China

**Keywords:** *Arthrinium phaeopermum*, disease resistance gene, *CCoAOMT2*, *CAD5*, RNAi

## Abstract

*Bambusa pervariabilis × Dendrocalamopsis grandis* shoot blight caused by *Arthrinium phaeospermum* is a fungal disease that has affected a large area in China in recent years. However, it is not clear which genes are responsible for the disease resistance of *B. pervariabilis* × *D. grandis*. Based on the analysis of transcriptome and proteome data, two genes, *CCoAOMT2* and *CAD5*, which may be involved in disease resistance, were screened. Two gene expression-interfering varieties, COF RNAi and CAD RNAi were successfully obtained using RNAi technology. Quantitative real-time fluorescence (qRT-PCR) results showed that *CCoAOMT2* gene, *CAD5* gene and seven related genes expression was down-regulated in the transformed varieties. After inoculating pathogen spore suspension, the incidence and disease index of cof-RNAi and cad-RNAi transformed plants increased significantly. At the same time, it was found that the content of total lignin and flavonoids in the two transformed varieties were significantly lower than that of the wild-type. The subcellular localization results showed that both CCoAOMT2 and CAD5 were localized in the nucleus and cytoplasm. The above results confirm that the *CCoAOMT2* and *CAD5* genes are involved in the resistance of *B. pervariabilis × D.*
*grandis* to shoot blight through regulating the synthesis of lignin and flavonoids.

## 1. Introduction

*Bambusa pervariabilis × Dendrocalamopsis grandis* is the product of the female parent *B. pervariabilis* and the male parent *D. grandis*. *B. pervariabilis* × *D. grandis* is widely used in papermaking and in returning farmland to forest, which also promotes the construction of ecological barriers in the middle and lower reaches of the Yangtze River. However, in recent years, it has been found that *B. pervariabilis × D. grandi**s* shoot blight caused by *Arthrinium phaeospermum* (Corda) M.B. Ellis has caused damage to more than 3000 hm^2^, leading to large economic losses [1] and greatly threatening the process of returning farmland to forest and the construction of ecological barriers in the Yangtze River Basin. At present, in addition to physiological research on the disease, molecular biology research has also made some progress. However, there is a lack of research on the molecular mechanism of the disease resistance of *B. pervariabilis* × *D. grandis*.

Disease resistance breeding is an effective method to improve the disease resistance of forest trees. It has the advantages of good economic benefits and environmental friendliness. The conventional methods of disease resistance breeding are systematic breeding and cross-breeding, but the breeding period is long. In recent years, with the development of molecular biology and tissue culture technology, bioengineering methods have been gradually applied to disease resistance breeding. Breeding resistant cultivars using resistance genes is currently the most effective and economical strategy for controlling crop diseases [2]. However, in bamboo plants, especially in *B. pervariabilis* × *D. grandis*, this kind of research is very few. Our previous studies have shown that many genes involved in cell wall composition synthesis, redox reactions, signal transduction and synthesis of secondary metabolites were significantly different in *B. pervariabilis* × *D. grandis* after pathogen infection. Twenty-one candidate genes for blight resistance, such as *pme53*, *CCoAOMT2*, *cad5*, *pod*, *gdsl-ll* and *Myb4l*, were found [3,4]. In higher plants, the intermediate products produced by the glycolysis pathway and the pentose phosphate pathway can be converted to phenylalanine through the shikimic acid pathway [5]. Phenylalanine, as the initial reaction substrate of the phenylpropanoid biosynthesis pathway in plants, can finally synthesize intermediate products such as ferulic acid, cinnamic acid and chlorogenic acid through the phenylpropanoid metabolism pathway, and then produce lignin, flavonoids, isoflavones and other substances [6]. Phenylpropane compounds are important components of plant defense in plants. Changes in many phenylpropane compounds and related enzymes involved in the synthesis of phenylpropane compounds are widely involved in plants’ responses to biological and abiotic stresses in the process of plant growth and development, and help plants resist external physical and biological invasion [7,8,9]. Both CCoAOMT (caffeoyl coenzyme, a 3-O-methyltransferase) and CAD (Cinnamyl alcohol deaminase) are important enzymes in the phenylpropanoid biosynthesis pathway in plants (Appendix A). Studies have shown that the *CCoAOMT* gene and the *CAD* gene are not only involved in lignin and flavonoid synthesis in the phenylpropanoid biosynthesis pathway but also related to plant resistance [10,11]. When plants are under the conditions of drought [12], low temperature [12,13], salt stress [14,15], hormone induction [15] and pathogen infection [16,17], CCoAOMT and CAD participate in the mechanisms of plant stress responses. However, research on the *CCoAOMT* gene and the *CAD* gene in bamboo generally focuses on gene cloning and expression analysis. There are few studies on the function of the *CCoAOMT* gene and the *CAD* gene in bamboo, although there is a report on the function of the *CAD* gene in *Bambusa multiplex* [18]. In particular, there is no report on the function of the *CCoAOMT* and the *CAD* genes in *B. pervariabilis* × *D. grandis*.

The methods of gene function identification are generally divided into transgenic technology, gene knockout technology and gene silencing technology. RNAi (RNA interference), also known as post-transcriptional gene silencing (PTGS), is characterized by specificity. An advantage of RNAi is that it can accurately identify and cut the mRNA of the corresponding gene, but has no interference with irrelevant genes, and has a significant interference effect on the target gene, which also confirms the signal amplification effect in the process of RNA interference [19,20,21,22,23,24,25]. In plants, using RNAi, dsRNA can be designed and synthesized and introduced into plants. Through the RNA interference mechanism in plants, the key pathogenic genes of pathogens can be silenced, so as to improve disease resistance, some examples of it are the resistance by RNAi against some viruses, fungi, pests, nematodes, parasitic weeds [26,27,28,29], etc.

In this study, the disease-resistant candidate genes of *B. pervariabilis* × *D. grandis* were screened by early transcriptome and proteome sequencing technology. On this basis, RNAi technology was used to interfere with the expression of disease-resistant candidate genes of *B. pervariabilis* × *D. grandis*, so as to verify the function of disease-resistant genes of *B. pervariabilis* × *D. grandis*, and finally to determine the main disease-resistant genes and analyze the internal causes of resistance to shoot blight of *B. pervariabilis* × *D. grandis* on the molecular level. This study provides a basis for in-depth research on the disease-resistance mechanism of *B. pervariabilis* × *D. grandis* and the molecular mechanism of pathogen–host interaction, as well as the development of new strategies for sustainable and effective control of forest branch and stem diseases.

## 2. Results

### 2.1. Gene Sequence Analysis and Protein Structure Prediction

Using the NCBI online blast nucleic acid sequence analysis tool, the results from the database were compared. They showed that the CDS sequences of *CCoAOMT2* and *CAD5* genes are 100% similar to the genes annotated with the agreed function of *Phyllostachys edulis* (FP094556.1) and *Bambusa multiplex* (GU9855522.1). According to the comparison results, the annotation of different families, genera and model plants were selected as the phase protein gene sequence using MEGA 5.1. The phylogenetic tree was constructed using the neighbor-joining algorithm (Figure 1). MEME was used to analyze the conserved sequences of the *CCoAOMT2* and *CAD5* genes. According to the principle of E-value < 0.05, 18 and 20 important domains of *CCoAOMT2* and *CAD5* genes were predicted, respectively (Figure 2). The amino acid residues of the *CCoAOMT2* and *CAD5* genes were 248 and 368, respectively, the total average hydrophilicity was −0.238 and 0.015, respectively, the theoretical isoelectric points were 5.21 and 6.06, respectively, and the predicted molecular weights were 27.84 kd and 39.47 kd, respectively. The proteins translated by the *CCoAOMT**2* and *CAD5* genes were modeled by SWISS-MODEL. The comparison results showed that the proteins that most matched CCoAOMT2 and CAD5 in the database were 5 kva.1.A, which was annotated as sorghum caffeyl-Coenzyme A methyltransferase, and 2cf5.1. B, which was annotated as Arabidopsis cinnamyl dehydrogenase. The consistency is higher than 70%, and its function can be preliminarily judged to be consistent (Figure 3).

### 2.2. B. pervariabilis × D. grands Total RNA Extraction and Reverse Transcription

RNA mass was detected using 1% agarose gel electrophoresis (Appendix A). The two bands of 28S and 18S from top to bottom are clearly visible in the figure, and although the 5.8S was slightly dispersed, it did not affect the follow-up experiment. The RNA extraction was complete, and the RNA extraction product can be used for subsequent experiments. The reverse transcription kit of Trans Biotechnology Co., Ltd. (Beijing, China) was used to reverse the total RNA of *B. pervariabilis × D. grands* to cDNA.

### 2.3. Construction and Transformation of RNAi Expression Plasmids of CCoAOMT2 and CAD5 Genes

Using the pHANNIBAL vector as the intermediate vector, the forward and reverse target fragments were inserted into both ends of the intron sequence of the pHANNIBAL vector, respectively (Appendix A), and then the complete fragment was obtained by enzyme digestion, as shown in Figure 4a. The pCAMBIA1301-35SN expression vector was constructed to interfere with the expression of two candidate disease resistance genes (CCoAMT2, CAD5). The positive colonies obtained after *E. coli *transformation sequencing with Zf/r-COF and Zf/r-CAD primers are shown in Figure 4b. The correctly sequenced recombinant plasmid DNA was transferred into Agrobacterium tumefaciens, and transgenic *B. pervariabilis* × *D. grandis* was cultivated according to the methods of Yuan [30]. The tissue culture process is shown in Figure 5. The positive plants were screened by PCR with hygromycin primer hyg501 f/R (Appendix A). The results are shown in Figure 4b.

### 2.4. Expression Analysis of CCoAOMT2 Gene and CAD5 Gene

The expression of the *CCoAOMT2* gene in the cof-RNAi transformed strain after 8 weeks of transplanting was significantly lowered compared with that of wild-type *B. pervariabilis* × *D. grandis* (Figure 6). This shows that the expression of the *CCoAOMT2* gene in *B. pervariabilis* × *D. grand* was inhibited after the transfer of the *CCoAOMT2* gene conserved region fragments of sense fragments and antisense fragments, and the expression of the *CAD5* gene of the cad-RNAi transformed plant after 8 weeks of transplanting was also significantly lowered compared with wild-type *B. pervariabilis* × *D. grandis*, compared with wild-type *B. pervariabilis* × *D. grandis*, indicating that the expression of *CAD5* genes in *B. pervariabilis* × *D. grandis* was significantly inhibited after the transgression of the gene conserved region of CAD5 gene fragments of sense fragments and antisense fragments.

### 2.5. Disease Resistance Level of Transgenic Plants

Taking the wild-type *B. pervariabilis* × *D. grandis* as the control, it was transplanted 3 months later with *A. Phaeospermum* spore suspension (10^6^ cfu/mL) and treated with cof-RNAi, cad-RNAi, and the wild-type *B. pervariabilis* × *D. grandis*, respectively. The incidence rate was calculated after 25 days of inoculation and the disease index was calculated (Table 1). The incidence of the wild-type *B. pervariabilis* × *D. grandis* was the lowest in the statistical control group, which was 16.7 ± 4.7%, and the lowest in the three groups of the wild-type *B. pervariabilis* × *D. grandis* disease index, which was 9.2 ± 3.1. The *CCoAOMT2* gene RNAi interferer strain cof-RNAi had the highest incidence of 86.7 ± 4.7%, which was significantly different from the incidence of the control group, and the disease index was higher, 42.5 ± 2.0, which was significantly different from the control group. The incidence of the *CAD5* gene RNAi interferer strain cad-RNAi was the second-highest, at 83.3 ± 9.4%, which was significantly different from that of the control group, and the disease index was 55.8 ± 5.1, which was significantly higher than the disease index of the control group. There were no significant differences in the incidence and disease index between the *CCoAOMT2* gene RNAi strain cof-RNAi and the *CAD5* gene RNAi interferer cad-RNAi.

After 25 days of inoculation, on a small number of the leaves of wild-type *B. pervariabilis* × *D. grandis* there appeared a small number of disease spots, the color of the branches was a healthy green, the dead branch phenomenon was not observed, and there were new tillers growing. After cof-RNAi and cad-RNAi inoculation for 25 days, spots appeared on more than 75% of the leaves, and some of the leaf spots covered the entire leaf, resulting in yellow wilt on the leaves, and some leaves finally falling off; further, brown spots appeared on the branches, and the branches lost water and yellowed (Figure 7). At the same time, there were obvious differences in the plant height, tillering number, leaf size and other indicators between wild-type *B. pervariabilis* × *D. grandis*, the cof-RNAi transformation plant, and the cad-RNAi transformation plant. The wild-type *B. pervariabilis* × *D. grandis* plants were taller, had more tillers, and their leaf size was significantly larger than that of cof-RNAi and cad-RNAi.

### 2.6. Total Lignin Analysis and Total Flavonoids

#### 2.6.1. Total Lignin Analysis

Lignin content in the stem tissues of wild-type *B. pervariabilis* × *D. grandis*, cof-RNAi and cad-RNAi were determined by lignin spectrophotometry (Table 2). The wild-type *B. pervariabilis* × *D. grandis* lignin content was the highest, at 4.064 mg/g, while the cof-RNAi transform plant had the lowest lignin content of 2.727 mg/g, thus recording a decrease of 33%, and the lignin content of the cad-RNAi transform plant was slightly higher, at 3.152 mg/g, thus recording a decrease of 22%. Overall, compared with the lignin content of wild-type *B. pervariabilis* × *D. grandis*, the lignin content of the RNAi transforming plant and the cad-RNAi transforming plant was not significant.

#### 2.6.2. Total Flavonoids Analysis

Flavonoid content in the stem tissues of wild-type *B. pervariabilis* × *D. grandis*, cof-RNAi and cad-RNAi were determined by spectrophotometry (Table 3). The wild-type *B. pervariabilis* × *D. grandis* flavonoids content was the highest, at 5.63 mg/g, while the cad-RNAi transform plant had the lowest flavonoids content of 3.01 mg/g, and the flavonoids content of the cof-RNAi transform plant was slightly higher, at 3.84 mg/g. Overall, compared with the flavonoid content of wild-type *B. pervariabilis* × *D. grandis*, the flavonoid content of the RNAi transforming plant and the cad-RNAi transforming plant was not significant.

#### 2.6.3. Correlation Analysis

Correlation analysis was performed on the relationships between the gene expression, lignin content and flavonoid content and the incidence, disease index, lignin content data and flavonoid content, respectively (Table 4). The correlation between the expression of the gene of interest and the incidence, disease index, lignin content, and flavonoid content and the correlation between the lignin content, flavonoid content and the incidence and disease indexes were significant. It was noted that there was no significant correlation between lignin content and flavonoid content. The above results show that *CCoAOMT2* and *CAD5* are involved in the synthesis of lignin and flavonoids, which plays a role in the defense mechanism of *B. pervariabilis* × *D. grandis* against shoot blight.

### 2.7. Gene Expression Analysis of Transformants

Using the STRING11.5 protein interplay database, *CCoAOMT2* and *CAD5* were analyzed, and seven interplay proteins were screened according to the transcriptome annotation information data of *B. pervariabilis* × *D. grandis*: 4-coumarate CoA ligase isoform 2 (4cl. 2), hydroxycinnamoyltransferase, cinnamoyl CoA reductase, cytosolic aldehyde dehydrogenase RF2D, glycosyltransferase, peroxidase 4, and peroxidase.

To verify the expression of the seven interplay proteins at the transcription level, seven pairs of primers were designed using the software Oligo V. 7, Molecular Biology Insights, Inc. (Cascade, CO, USA) (Appendix A). In the cof-RNAi transform strain, the expression of seven related proteins showed a certain downregulation at the transcription level compared with that of wild-type *B. pervariabilis* × *D. grandis* (Figure 8), of which hydroxycinnamoyl transferase expression was the most significantly downregulated, followed by Glycosyltransferase, cinnamoyl CoA reductase, cytosolic aldehyde dehydrogenase RF2D and 4-coumarate CoA ligase, and the relative expression of the remaining two genes was slightly downregulated, of which peroxidase 4 had the lowest decrease in relative expression, which was not significantly different from the control group. In the cad-RNAi transform strain, the expression of seven related proteins in the cad-RNAi transforming plant showed a certain downregulation compared with the expression of wild-type *B. pervariabilis* × *D. grandis*, of which hydroxycinnamoyl transferase was the most significantly downregulated, followed by 4-coumarate CoA ligase, cinnamoyl CoA reductase and glycosyltransferase, and the remaining three genes were slightly and relatively downregulated. Among them, the expression of peroxidase 4 was consistent with the expression of wild-type *B. pervariabilis* × *D. grandis* in the control group: there was no upregulation or downregulation, and there was almost no difference.

### 2.8. Subcellular Localization of CCoAOMT2 Gene and CAD5 Gene

After the sequence verification of the complete coding region gene amplification fragments of *CCoAOMT2* and *CAD5* were correct, the gel of interest was recovered and the recovered fragments were electrophoretically detected on a 1% agarose gel (Appendix A). The figure was marked as 1 where the lane was the complete CDS region of *CCoAOMT2*, and the enzyme cutting site of 6 bp and the protective base of 2 bp were attached at each end of the fragment, leading to a total of 763 bp. Further, the 2 lane was marked as the complete CDS region of *CAD5*, and the enzymatic resination site and 2 bp of the protective base were attached at each end of the fragment, leading to a total of 1124 bp. From the figure it can be seen that the *CCoAOMT2* recovered fragment was located between 750 bp and 1000 bp of the 2Kmarker, and the *CAD5* recovered fragment was located between 1000 bp and 2000 bp of the 2Kmarker indicating that the amplification product was correct; this can be used for the next experiment. *CCoAOMT2* and *CAD5* gene amplification products and pCAMBIA1300-EGFP-MCS plasmids were transferred to Agrobacterium EHA105 after double-cutting recovery by KpnI. and XbaI., respectively, and the fusion plasmids were named cof-EGFP and cad-EGFP, respectively. After 2–3 d of incubation on YEP plates, positive colonies were selected to expand the culture, and these were injected into the back of Tobacco Ben’s leaves for 1 month of incubation. After 48–72 h of incubation under suitable conditions, images were recorded by laser confocal microscopy imaging, and the results found that the no-loaded tobacco fluorescence signal of the positive control conversion pCAMBIA1300-EGFP-MCS appeared randomly in the whole plant cells, and at the same time transferred to the markers localized to the membrane and nucleus, respectively. Further, the tobacco localization infected with cof-EGFP and cad-EGFP fusion plasmid Agrobacterium was found to be similar (Figure 9). The fusion proteins cof-EGFP and cad-EGFP coincided with the nuclear localization marker fluorescence signal, but not with the membrane localization marker fluorescence signal, indicating that CCoAOMT2 and CAD5 are localized to the nucleus and cytoplasm.

## 3. Discussion

In this experiment, the clone sequences of the two genes of interest were highly similar to the genes annotated as CCoAOMT and CAD in the database, and most of these genes came from bamboo; in particular, the most similar sequences were from bamboo, which was consistent with the results of previous studies [31,32]. In a conservative regional analysis of the sequence, both gene sequences were found to be very conserved, and the identified conserved motif covered almost the entire sequence. In particular, an S-adenosyl methionine binding site, which is its functional domain [33], was identified in the *CCoAOMT2* gene, which initially determined that the gene expression product could perform methyl transfer functions. This result is consistent with that found for *Eucalvptus urophylla* [34] and *Dendrocalamus sinicus* [31]. In the protein tertiary structure analysis, the model results matched by the comparison of the sequences showed that the two sequences matched the highest degree of caffeyl-CoA methyltransferase and cinnamyl alcohol dehydrogenase, respectively, and the consistency reached more than 70%, meaning that the model can be considered to have fully represented the real results.

The phenomenon of RNAi is widely confirmed in plants [35,36,37], where there are two different interfering mechanisms at the molecular level; one is the siRNA-producing pathway, and the other is the miRNA-producing pathway [38]. In this experiment, by constructing a ds-RNA transgenic plasmid, the *B. pervariabilis* × *D. grandis* transformed strain was successfully obtained. The expression of the *CCoAOMT2* and *CAD5* genes of interest was analyzed by q-PCR technology, and the expression of both genes in the transformed strain was significantly inhibited. The results showed that there was an RNA interference mechanism through the siRNA pathway in *B. pervariabilis* × *D. grandis*, which was similar to that found in carnation [39].

In the study of anti-blight disease in melon, it was found that *CmCAD2* and *CmCAD5* played a certain anti-disease role. In this experiment, the incidence of *CCoAOMT2* and *CAD5* gene expression was significantly higher than that of the control group, reaching 86.7%, 83.3%, 70.0% and 66.6% higher than that of the control group, respectively. At the same time, the disease index was also significantly increased to 42.5 and 55.8, respectively, which were higher than the control group values of 33.3 and 46.6, respectively. These results were similar to the previous research results [16], which also showed that *CCoAOMT2* and *CAD5* genes were involved in *B. pervariabilis* × *D. grandis*’s resistance to the shoot blight caused by the dark spore node rhombus. It was found that the *AtCAD4* and *AtCAD5* genes in the *AtCAD* gene family of Arabidopsis thaliana were significantly expressed after 6 h of Phytophthycosis infestation with Arabidopsis [40], while the other genes were not significantly different in the family, and the *AtCAD4* and *AtCAD5* genes were closely related to plant resistance in the *CAD* gene family. In the present experiment, the disease index of the *CAD5* gene expression interferer was higher than that of the *CCoAOMT2* gene expression interferer, which was consistent with the above-mentioned result. Due to the interference of *CAD5* gene expression, which is more significantly related to disease resistance, plant disease resistance is reduced, showing a higher disease index. The *CCoAOMT2* gene was more sensitive to the RNAi mechanism, so it recorded a higher incidence in *B. pervariabilis* × *D. grandis* populations.

Lignin biosynthesis is mainly regulated by three functional enzymes, namely PAL, C4H, and 4CL in the amphetamine pathway, COMT, CCoAOMT, and F5H associated with monomer synthases, and CAD and CCR [41], which are downstream regulatory enzymes for lignin synthesis. In this experiment, the lignin content of the plants inhibited by expression inhibition of the two genes was determined by interfering with the expression of the lignin biosynthetic enzymes CCoAOMT and CAD, respectively, and it was found that the lignin content of the transformed plant decreased to varying degrees [42,43,44,45,46,47], especially the *CCoAOMT* expression interference transform strain. This may have been due to the fact that the expression of the *CCoAOMT2* gene is more sensitive to RNA interference. This result is consistent with the more significant decrease recorded in the expression of the *CCoAOMT2* gene of interest. In general, after the interference in *CCoAOMT2* and *CAD5* gene expression, the lignin content decreased significantly compared with the control group, which proved that the *CCoAOMT2* and *CAD5* genes were involved in regulating the synthesis of lignin in *B. pervariabilis* × *D. grandis*, which was consistent with the previous research results. Flavonoids are important substances for plants to resist stress under stress and have a variety of bioactive functions. In this experiment, it was found that the content of flavonoids decreased significantly, and the incidence rate and disease index of *B. pervariabilis* × *D. grandis* also increased significantly. It shows that flavonoids are related to the resistance of *B. pervariabilis* × *D. grandis* to shoot blight.

Hydroxycinnamoyl transferase (HCT) belongs to the plant acyltransferase family (BAHD family), which is able to catalyze the conversion of acyl-coenzyme A into a variety of downstream products [48]. In *Arabidopsis*, it was found that lignin synthesis was inhibited, and plant dwarfing was evident after interference with the expression of quinic acid/shikimic acid-hydroxycinnamyltransferase (HCT), suggesting that hydroxycinnamyltransferase was associated with lignin synthesis in plants [49]. It was also found that HCT has the characteristics of the broad spectrum of the reaction substrate, which can catalyze the acylation of a variety of substrates, including cinnamoyl CoA, p-coumarincoa CoA, caffeanoyl CoA, feruloyl CoA, and mustardyl CoA [50]. In this experiment, the expression of the *CCoAOMT2* and *CAD5* genes related to lignin synthesis was interfered with, resulting in blocked caffeyl CoA methylation and oxidation resistance of hydroxycinnamaldehyde in lignin synthesis, which led to the decrease in lignin content, while hydroxycinnamoyltransferase was positively correlated with lignin synthesis. Therefore, the expression of hydroxycinnamoyltransferase, a key enzyme of lignin synthesis, was downregulated after the interference of the expression of *CCoAOMT2* and *CAD5* genes.

The subcellular localization of the CCoAOMT2 and CAD5 proteins was predicted using a database prior to the trial, showing that the CCoAOMT2 and CAD5 proteins were most likely to be localized to the cytoplasm, but it was also observed that the localization results of the two genes were different in different plants [51,52,53,54]. The CDS cloning of the two genes in this experiment was then connected to a plasmid with EGFP for fusion expression. The results showed that CCoAOMT2 and CAD5 were localized to the cell membrane and nucleus, which was generally consistent with the prediction results, and there was no nucleus localization in the prediction results of CAD5. Therefore, the second fusion expression was tried, and the experimental results were consistent with the first experimental results. The two experiments thus proved the credibility of the experimental results. Previous results found for Kuerle Pear [55], in which CCoAOMT and CAD localization in the cytoplasm were observed, are consistent with the results of this experiment. However, it was noted that CCoAOMT2 and CAD5 were also located in the nucleus in addition to the cytoplasm, which was different from the above results; further, according to Liu [56], who studied the subcellular localization of the two metabolic enzymes of pear lignin, CCoAOMT2 and CAD5 were found to have the function of the transcription factor and could bind to DNA in the nucleus to regulate gene expression, so they were also located in the nucleus.

In conclusion, *CCoAOMT2* and *CAD5* may be involved in the resistance of *B. pervariabilis × D.grandis* to shoot blight as disease resistance genes. However, the interaction between target genes and pathogens at the protein level is not clear. In the future, we can continue to deeply explore the disease-resistance mechanism of CCoAOMT2 and CAD5 at the protein level.

## 4. Materials and Methods

### 4.1. Materials

*Bambusa pervariabilis × Dendrocalamopsis grandis* seedlings were obtained from the forest protection laboratory of Sichuan Agricultural University, China. *A. phaeospermum* was isolated by tissue isolation method [57] from diseased *B. pervariabilis* × *D. grandis*, the accession number of *A. phaeospermum* ITS in NCBI database is OK626768. The isolate was stored in China Forestry Culture Collection Center, numbered cfcc 86860 (http://www.cfcc-caf.org.cn/, accessed on 6 April 2007).

### 4.2. Methods

#### 4.2.1. Gene Sequence Analysis, Protein Structure Prediction

The candidate gene transcriptome sequences were aligned in the NCBI database, and the phylogenetic tree was constructed using MEGA5.1’s neighbor-joining algorithm [58]. The conserved domains of proteins were predicted by MEME [59]. The isoelectric points of proteins were analyzed by EXPASY [60]. The tertiary structures of proteins were analyzed by SWISS-MODEL [61].

#### 4.2.2. Total RNA Extraction and cDNA Synthesis of *B. pervariabilis* × *D. grandis*

Fresh shoots of *B. pervariabilis* × *D. grandis* were ground in liquid nitrogen, and total RNA was extracted from tissue cells using the PlantTrol RNA extraction kit (TransGen, TransGen Biotech Co., Beijing, China). The concentration and integrity of RNA were determined by Nanodrop microspectrophotometer (NANODROP, ThermoFisher Scientific-CN, Shanghai, China) and agarose gel electrophoresis (DYY-6D, LIUYI, Beijing, China), respectively. EasyScript^®^ All-in-One First-Strand cDNA Synthesis SuperMix (TransGen) of TransGen Biotech Co. China, was used for PCR, reverse transcription into cDNA was carried out, and the results were kept on standby at −20 °C.

#### 4.2.3. Construction and Transformation of RNAi Expression Plasmids of *CCoAOMT2* and *CAD5* Genes

The recombinant primers Zf/r-COF, Ff/r-COF, Zf/r-CAD and Ff/r-CAD (Appendix A) were designed by software PRIMER PREMIER V. 5.0, Premier Biosoft International (Palo Alto, CA, USA). Reaction system: cDNA 1 µL, F/R 1/1 µL, 2× TransTaq HiFi PCR SuperMix (TransGen) 25 µL nuclease-free water 22 µL. Reaction procedure: 94 °C, 5 min (94 °C 30 s, 55 °C 30 s, 72 °C 2 min), 34 cycles, 72 °C, 10 min. PCR products were detected by 1% agarose gel electrophoresis and recovered by gel cutting. The linearization vector pHANNIBAL was connected to the forward and reverse fragments by clonexpress one-step cloning kit. Reaction system: linearization vector pHANNIBAL 5 μL, forward or reverse fragment DNA 3 μL, 5× CE Buffer 4 μL, Exnase 2 μL, ddH_2_O 6 μL. After reaction at 37 °C for 30 min, the recombination reaction was completed by cooling at 4 °C for 5 min. The connecting products were transformed into competent cells of *E. coli* stbl3. Monoclonal antibodies were selected for expanded culture, and then the plasmids were extracted. The recombinant plasmid pHANNIBAL and pCAMBIA1303-35s were treated with XbaI, and KpnI performed double enzyme digestion. Reaction system: plasmid DNA 10 μL, 10× FlyCut Buffer 5 μL, XbaI 1 μL, KpnI 1 μL, dd H_2_O 33 μL. Next, 15 min at 37 °C and 20 min at 65 °C after mixing, agarose gel electrophoresis was carried out after gel recovery. T4 ligase (Taraka, Takara Biomedical Technology (Beijing) Co., Ltd., Beijing, China) was used to connect the target fragment and pcambia1301-35s linearized plasmid overnight at 16 °C.

#### 4.2.4. Agrobacterium Transformation and *B. pervariabilis* × *D. grandis* Tissue Culture

Transformation of *E. coli* DH5α took place using ligation product α Competent cells. The recombinant plasmid was introduced into *Agrobacterium* EHA105 competent cells by freeze–thaw method, and the agrobacterium that was identified as positive by PCR was impregnated with *B. pervariabilis* × *D. grandis* embryogenic callus [29]. The basic medium and plant hormone conditions used in the process of tissue culture were as follows: MS medium was added with 4 mg/L 2,4-D for callus induction, MS basic medium was added with 3 mg/L 6-BA and 3 mg/L KT for callus differentiation, MS basic medium was added with 2 mg/L NAA for rooting. Primer hyg501-f/r was designed for the positive transformants screening by PCR amplification of hygromycin resistance genes. The transformed strains with positive *CCoAOMT2* and *CAD5* gene expression interference were named cof-RNAi and cad-RNAi, respectively.

#### 4.2.5. Quantitative Detection of Gene Expression of *CCoAOMT2* and *CAD5* in RNAi Positive Transformed Vaccine by q-PCR

RNA from wild-type *B. pervariabilis* × *D. grandis* and cof-RNAi and cad-RNAi was extracted and reverse transcribed into cDNA. Using the cDNA as the template, primers were designed from the specific fragments of *CCoAOMT2* and *CAD5* genes in *B. pervariabilis* × *D. grandis*, and real-time quantitative PCR (q-PCR) specific primers qCof-F/R, qCAD5-F/R were designed (Appendix A). The primers were synthesized by Tsingke Biotechnology Co., Ltd. (Beijing, China). GAPDH and Actin were used as internal reference genes. TransScript^®^ Green One-Step qRT-PCR SuperMix (TransGen) was used for qRT-PCR. The q-PCR system consisted of the following: 10 μL qPCR SuperMix, 0.4 μL RT/RI Enzyme Mix, 0.4 μL Passive Reference Dye, 7.4 μL ddH_2_O, 0.4 μL F/R, 1 μL RNA. The qPCR was performed: at 95 °C for 5 min, 94 °C for 30 s, 94 °C for 5 s, and 60 °C for 30 s. The third step and the fourth step were repeated 40 times. Each treatment group underwent qPCR three times, and the average value was calculated. Data were analyzed by the 2^−^^∆∆^^Ct^ method [62].

#### 4.2.6. Pathogen Inoculation and Disease Resistance Detection

The spores of *A. phaeospermum* cultured for 10 days were washed off with sterile water to prepare 10^5^ CFU/mL spore suspension. A total of 30 strains of cof-RNAi, 30 strains of 6 cad-RNAi and 30 wild-type seedlings were selected, and these were inoculated with spore suspension by acupuncture method [63]. Each plant was inoculated with 3 nodal forks, and each place was inoculated with 5 mL spore suspension, which was kept wet for 12 h, and sterile water was used as the control. The incidence rate and disease index were calculated after 25 days of inoculation.
 I(%)= D/T ×100
where *I*, D and T are the incidence, the number of diseased plants and the number of total plants, respectively.

The disease classification standard was as follows: 0: no wilt; Level 1: less than 25% withered branches; Level 2: 25–50% (including 25% and 50%) withered branches; Level 3: 50–75% withered branches (including 75%); Level 4: more than 75% withered branches [57].
DI=Σ (DN × L)/(TN ×M)×100
where *DI* is the disease index, DN is the disease level, L is the number of plants at each level, TN is the total number of diseased plants, and *M* is the maximum disease incidence.

#### 4.2.7. The Content of Total Lignin and Total Flavonoids

The content of lignin and flavonoids were determined by using the branches of the cof-RNAi, cad-RNAi transformed plants and wild-type *B. pervariabilis* × *D. grandis* as plant materials

The total amount of lignin was determined by the acetyl bromide method. The specific operation is to dry the sample at 80 °C to constant weight, crush it, sieve it through 40 mesh, weigh it about 5 mg (recorded as W) in 10 mL glass test tube (always use a glass test tube, not EP tube) and record it according to the instructions of Suzhou Keming Biotechnology Co., Ltd. (Suzhou, China), China lignin content kit (COMIN). The total amount of lignin can be calculated according to the standard curve and measurement data provided by the kit.
y=0.0694x+0.0068,R2=0.9889
Lignin (mg/g)=(ΔA−0.0068)÷0.0694×V(mL)×10−3÷W(g)×T=0.0294×(ΔA−0.0068) ÷W(g)×T
where *V* is the total reaction volume of 2.04 mL, *W* is the sample mass, and *T* is the dilution multiple.

The principle is that flavonoids can form a stable complex with aluminum salt and obtain a stable characteristic absorption peak in the visible spectrum. The content of total flavonoids was determined according to the method of Liu et al. [64]. 0.5 g of fresh tissue of the stem was placed in 70% ethanol for ultrasonic treatment at 35 °C for 45 min, and its concentration was measured by colorimetry; Add 1 mL of solution and 5 mL of deionized water to a 25 mL test tube, add 1 mL of 5% (mass volume fraction) NaNO_2_ to the reaction mixture and keep it at room temperature for 6 min; Then add 1 mL of 10% (mass volume fraction) AlCl_3_ · 6H_2_O and keep it at room temperature for 6 min; Then add 1 mL of 1 mol/L NaOH and fix the final volume to 25 mL with deionized water. The absorbance of the solution was immediately read at 512 nm.

The target gene expression, incidence rate, disease resistance index, lignin content and flavonoid content were measured according to the appeal method, and the software graphpad prism V. 8.0, GraphPad Software, LLC (San Diego, CA, USA) was used. Pearson correlation coefficient was used for correlation analysis.

#### 4.2.8. Expression Analysis of Related Genes

The STRING 11.5 (STRING: STRING: functional protein association networks (string-db.org)) protein interaction database was used to analyze CCoAOMT2 and CAD5, and then seven interacting proteins were screened from the annotation information data of *B. pervariabilis* × *D. grandis* transcriptome, which were 4-coumarate CoA ligase isoform 2 (4CL. 2), hydrocinnamoyltransferase, cinnamoyl CoA reductase, cytosolic aldehyde dehydrogenase RF2D, glycosyltransferase, cinnamoyl CoA reductase 2 and peroxidase. Taking the cDNA of wild-type *B. pervariabilis* × *D. grandis* as the control and GAPDH and Actin genes, two of the plant housekeeping genes, as the internal reference, the seven predicted interaction proteins were named 1–7 in the order of appeal, corresponding to primers 1-F/R to 7-F/R in Appendix A, and the fluorescence quantitative identification and analysis were carried out according to the method in 4.2.5.

#### 4.2.9. Subcellular Localization of *CCoAOMT2* and *CAD5* Gene

Primers CCoAOMT2-GFP-F/R and CAD5-GFP-F/R (Appendix A) were designed according to the transcriptome sequencing sequences of *CCoAOMT2* and *CAD5* genes. The coding sequences of *CCoAOMT2* and *CAD5* genes were amplified by PCR with the *B. pervariabilis × D. grand**is* cDNA as the template, and the PCR products were recovered. Kpn I and Xba I were selected as restriction endonuclease sites and connected by T4 DNA ligase. The recovered target fragments were connected with pCAMBIA1300-EGFP-MCS and then introduced into *E. coli* DH5α competent cells, which were screened for positive clones with primer yz-F/R (Appendix A) and sequenced. The correctly inserted plasmid was extracted and transferred into Agrobacterium EHA105 to infect tobacco [65]. After 48–72 h, observations were made under laser confocal microscope (FV3000, Olympus, Tokyo, Japan).

## Figures and Tables

**Figure 1 ijms-23-06760-f001:**
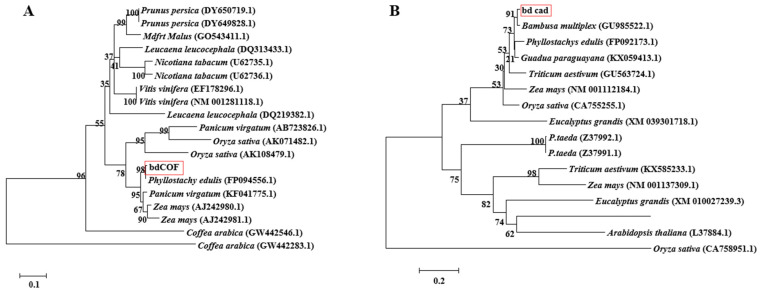
Target gene phylogenetic tree based on neighbor-joining algorithm. (**A**): *CCoAOMT2* phylogenetic tree; (**B**): *CAD5* phylogenetic tree. The red box highlights the gene of interest. The numbers above/below the nodes indicated bootstrap number.

**Figure 2 ijms-23-06760-f002:**
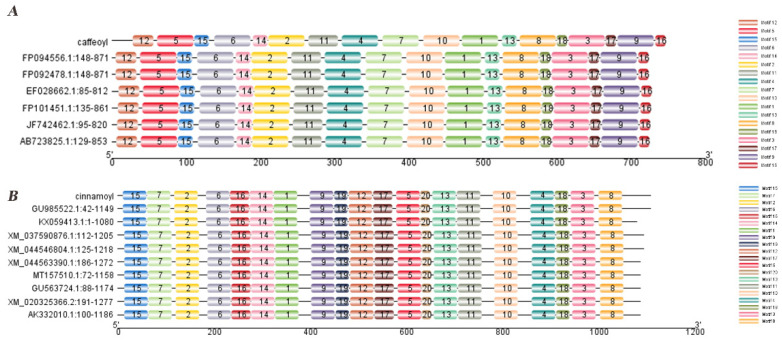
Sequence analysis of conserved motifs of target gene. (**A**): *CCoAOMT2* identified 18 conservative motifs, different motifs are distinguished by different color numbers; (**B**): *CAD5* identified 20 conservative motifs, different motifs are identified by different color numbers.

**Figure 3 ijms-23-06760-f003:**
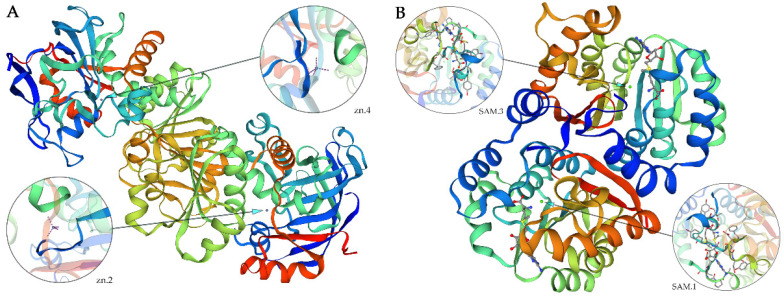
Homology modeling of CCoAOMT2 (**A**) and CAD5 (**B**) encoding proteins.

**Figure 4 ijms-23-06760-f004:**
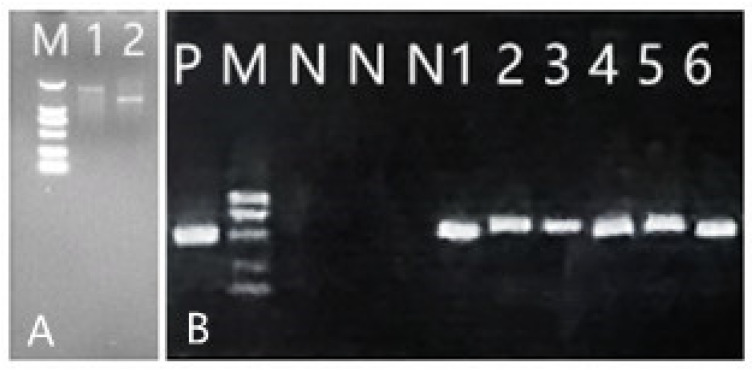
Electrophoresis diagram of RNA extraction from stem tip of *B. pervariabilis* × *D. grandis* (**A**); PCR of *Bambusa pervariabilis × Dendrocalamopsis grandis* transformed vaccine (**B**). M: DL2000 DNA marker; 1, 2, 3 and 4: *B. pervariabilis* × *D. grandis* RNA (**A**); M: 1000 DNA Marker (Fermentas); P: Positive control; N: Negative control; 1, 2, 3, 4, 5 and 6: PCR detection results of hyg501 fragment of transformed strain (**B**).

**Figure 5 ijms-23-06760-f005:**
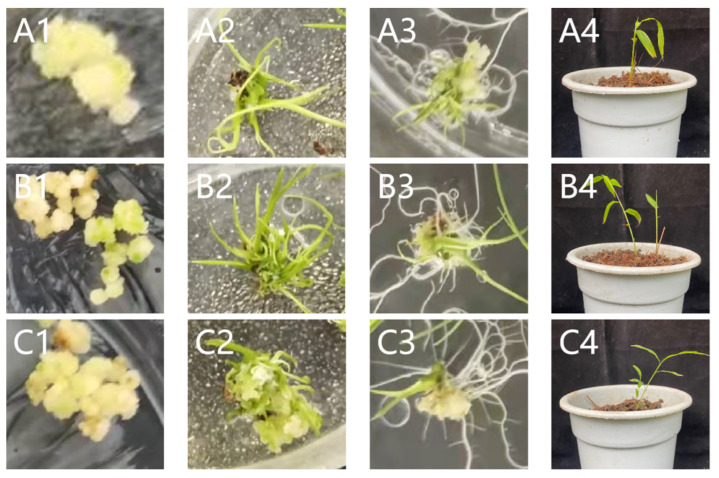
Genetic transformation of *B. pervariabilis* × *D. grandis*. (**A**): cof-RNAi; (**B**): cad-RNAi; (**C**): wild type *B. pervariabilis* × *D. grandis*; **1**: callus, **2**: differentiation of buds, **3**: differentiation of rooting, **4**: after transplanting.

**Figure 6 ijms-23-06760-f006:**
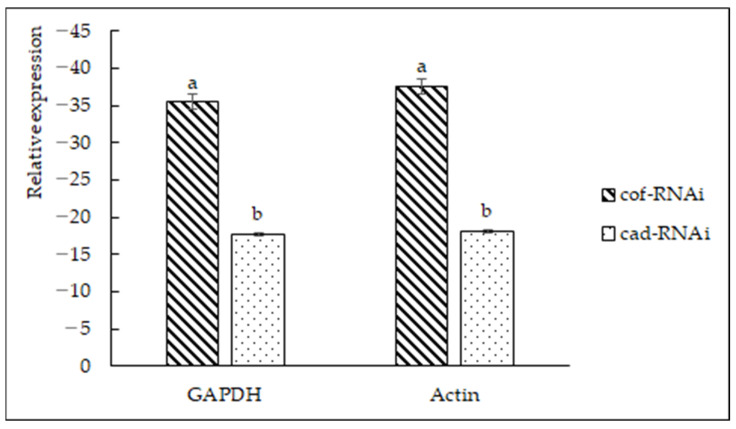
Relative expression levels of *CCoAOMT2* and *CAD5* genes. There are significant differences in the relative expression of different lowercase letters (*p* < 0.01).

**Figure 7 ijms-23-06760-f007:**
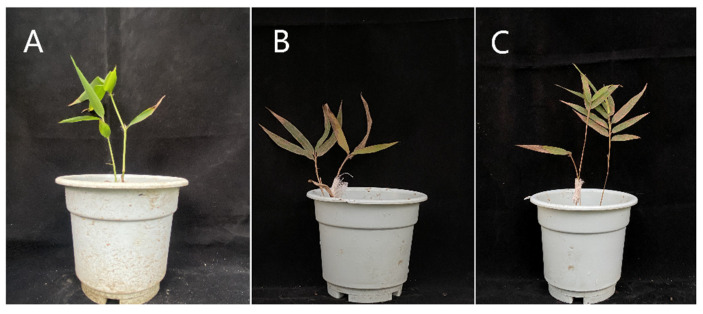
Symptom chart of different strains of 3-month-old *B. pervariabilis* × *D. grandis* after inoculation with pathogenic *A. phaeospermum*. (**A**): wild-type *B. pervariabilis* × *D. grandis*; (**B**): cof-RNAi, (**C**): cad-RNAi.

**Figure 8 ijms-23-06760-f008:**
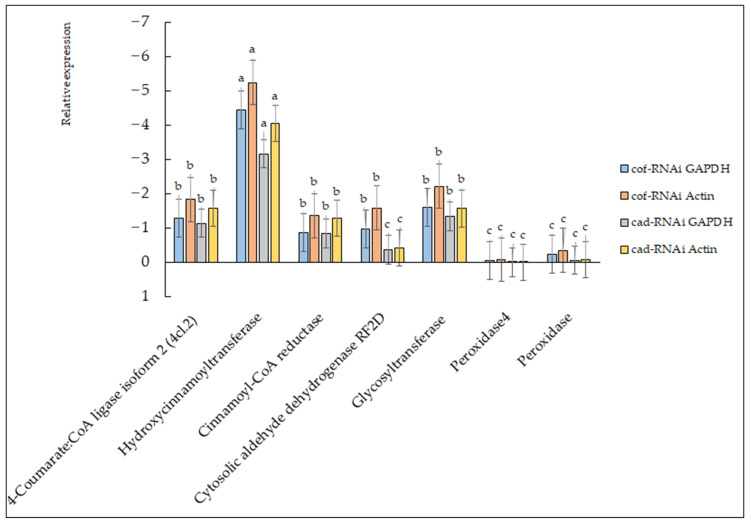
Changes in related protein gene expression levels. There are significant differences in the relative expression of different lowercase letters (*p* < 0.01).

**Figure 9 ijms-23-06760-f009:**
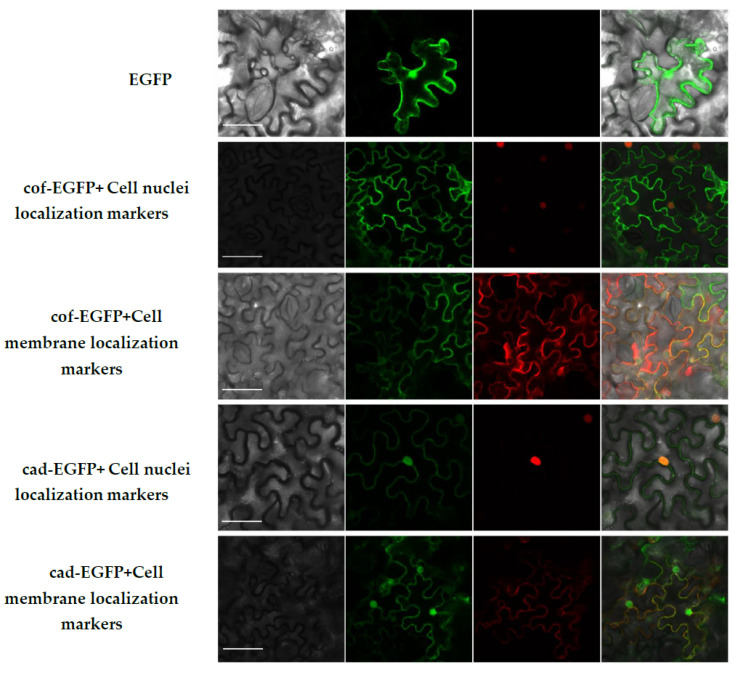
CCoAOMT2 and CAD5 subcellular localization. The white bar indicates 50 μm.

**Table 1 ijms-23-06760-t001:** Statistical table of the incidence rate and disease index of the three varieties.

Varieties	Incidence (%)	Disease Index
cof-RNAi	86.7 ± 4.7 ^a^	42.5 ± 2.0 ^a^
cad-RNAi	83.3 ± 9.4 ^a^	55.8 ± 5.1 ^a^
wild-type *B. pervariabilis* × *D. grandis*	16.7 ± 4.7 ^b^	9.2 ± 3.1 ^b^

The data are the average of three sets of repetitions; data in the same column followed by different lowercase letters indicate significant differences among different varieties after inoculation by the LSD test (*p* < 0.05).

**Table 2 ijms-23-06760-t002:** Lignin content in transgenic plants.

Varieties	Lignin Content mg/g
wild-type *B. pervariabilis* × *D. grandis*	4.064 ± 0.107 ^a^
cof-RNAi	2.727 ± 0.473 ^b^
cad-RNAi	3.152 ± 0.449 ^b^

The data are the average of three sets of repetitions; data in the same column followed by different lowercase letters indicate significant differences among different varieties after inoculation by the LSD test (*p* < 0.05).

**Table 3 ijms-23-06760-t003:** Flavonoid content in transgenic plants.

Varieties	Flavonoids Content mg/g
wild-type *B. pervariabilis* × *D. grandis*	5.63 ± 1.31 ^a^
cof-RNAi	3.84 ± 0.26 ^b^
cad-RNAi	3.01 ± 0.24 ^b^

The data are the average of three sets of repetitions; data in the same column followed by different lowercase letters indicate significant differences among different varieties after inoculation by the LSD test (*p* < 0.05).

**Table 4 ijms-23-06760-t004:** Correlation analysis of gene expression, incidence, disease index and lignin content.

	Incidence	Disease Index	Lignin Content	Flavonoids Content
*CCoAOMT2* relative expression	−0.9865 **	−0.9840 **	0.8849 *	0.9992 **
*CAD5* relative expression	−0.9810 **	−0.9877 **	0.8270 *	0.9991 **
lignin content	−0.7917 *	−0.7173 *	/	0.6515
flavonoids content	−0.9365 *	−0.9994 *	0.6515	/

*, indicating that the gene relative expression and lignin content of each target were significantly correlated with incidence, disease index and lignin content (*p* < 0.05, LSD test); **, indicating a very significant correlation (*p* < 0.01, LSD test).

## Data Availability

The raw datas have been published in the NCBI TSA datebase: GJJA00000000. The mass spectrometry proteomics data have been deposited to the ProteomeXchange Consortium [66] via the iProX partner repository [67] with the dataset identifier PXD014363, the subject ID is IPX0001646002.

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
