# Peer review of "Molecular Mechanisms of Phenylpropane-Synthesis-Related Genes Regulating the Shoot Blight Resistance of Bambusa pervariabilis × Dendrocalamopsis grandis"

_ijms, 2022, doi:10.3390/ijms23126760_

Round 1
Reviewer 1 Report
This manuscript is well described their methods and results. I'm happy to accept this manuscript. Check minor spells and grammar errors before published.
Reviewer 2 Report
The manuscript “Molecular mechanisms of phenylpropane-synthesis-related genes regulating the shoot blight resistance of Bambusa pervariabilis × Dendrocalamopsis grandis” deals with a fungal disease, the identification of the plant genes implied in the resistance against the pathogen, and the analyses of their function in the mechanisms of resistance.
The experimental design and analyses are adequate and detailed. The work carried out by the authors provides with a big amount of data that they exposed in detail, extracting abundant information from them. The manuscript is well written, detailed and exhaustive in procedures for analyses. It is well documented with figures and tables (included in the text, as well as supplementary material).
Some points should be corrected and others could be improved for clarification, as following:
- The authors should check the Abstract since it is too long, according to the Author’s guidelines. The abstract presented has 310 words, however it should not have more than 200 words in total. Since the authors repeat several times the scientific names of both, the pathogen and the plant species, maybe they could revise the writing style in order to try not to repeat so much both names or even using a common name if possible. Besides, they could try to present a shorter and more concise abstract when explaining the results.
- Regarding the Keywords, the authors do not need to include ‘Bambusa pervariabilis × Dendrocalamopsis grandis’ since it is already in the title. They could change it for another one (i.e. fungal pathogen, RNAi, etc.), increasing the chances of being found in bibliographic searches.
- The Introduction could be improved and be more informative if the authors give some information regarding the genetic resources of plant resistance available in that plant species against that disease. In the Introduction, the authors indicate that this fungal disease has caused big economic losses and that it is threatening some processes in the river basin. However, without any further explanation, they go directly to deal with the chemical pathways implied in resistance. It would be nice and informative if they could include a small paragraph (after line 48) dealing with the availability and use (or not) of resistance varieties, before going directly to resistance mechanisms.
- Lines 81-82 indicated some examples of the use of RNAi for plant resistance, so, please, write it that way in order to clarify the sentence. It could be written something like ‘some examples of it are the resistance by RNAi against some viruses, fungi….’.
- In Results, please check and correct the software’s name used (MEGA, line 101). Likewise, for line 109 (Swissmodel or SWISS-MODEL). Please, name all software in the same way in both, the Material and Methods, and the Results. Likewise, for the information about the kits used (name of company).
- Please, check and correct the name of vector in lines 133 and 134 (pHANNIBAL according to Material and methods).
- In Discussion, could the authors provide some references in line 311 related to those previous studies?
- Line 345, separate words in ‘above mentioned’.
- Lines 346-349: could you make the sentence clearer and without sentences into brackets?
- Lines 371: the scientific name of Arabidopsis should be in italics.
- Line 406: please write CCoAOMT2 in upper case, like in the rest of the text.
- In Materials and methods, please include some information about the fungi used in inoculations (line 411, ‘Materials’), even if some details (although not the source) is already included in lines 474-476.
- Lines 420 and 446: the scientific names should be in italics.
- Line 422-426: please check and complete the data of the RNA extraction kit used, the spectrophotometer and the agarose gel visualization. Some more data about PCRs carried out would be informative. Maybe the authors mean ‘TransGen Biotech Co, China. If so, please, complete it in lines 423 and 466.
- Line 470: could you specify the information about de standard curve that you name ‘dissolution curve’? It is advisable that the authors follow The MIQE Guidelines (Bustin et al., 2009) for qPCR experiments.
- Line 477: could the authors include a reference, if possible, related to the acupuncture method that it is used for inoculation with the pathogen?
- Section 4.2.7 (beginning in line491): it would be nice and informative if the authors include a couple of lines indicating the reasons for extracting lignin and flavonoids, as well as the samples used (the samples are supposed to be the plants inoculated before, and maybe the whole plant or only part of them). The authors mention them in the Abstract but not clearly in the Introduction and the Material and Methods. Please, include some information here.
- Line 496: the authors could complete the information about the kit used. Please, check if it is ‘Suzhou Keming Biotechnology Co., Ltd., China’, and include it if possible.
- Line 511: could the authors explain briefly the appeal method?
Reviewer 3 Report
Comments to the Author:
General comment:
This study focused on the molecular mechanisms of phenylpropane-synthesis-related genes regulating the shoot blight resistance of Bambusa pervariabilis × Dendrocalamopsis grandis. The data suggests that the CCoAOMT2 and CAD5 gene are involved in the resistance of B. pervariabilis × D. grandis to shoot blight through regulating the synthesis of lignin and flavonoids. The manuscript is well written, and the outcomes should serve as references for plant biologist and pathologists. I support the publication of this manuscript after proper corrections.
I have some minor comments that the authors should address.
Specific comments:
Abstract, Line 32: “CCoAOMT2 gene and CAD5 gene are involved in the resistance ...”; It should be written: CCoAOMT2 and CAD5 genes are involved in the resistance … Please remove the first “gene” and use plural.
Lines 70: same comment
Figure 1: indicate the bootstrap number.
Figure 2: Figure legend is overlapping with the scale. Please move the legend to the right.
Figure 8: Axes labels are too small, please improve size. Figure legend should be presented as a 4-line column in the right side of the graph. Color bars will probably improve the graph.
Figure9: “White ruler”. It should be white bar.
Check reference 29
References 41 and 45. “The Plant cell”. Cell should be in capital.
Round 2
Reviewer 2 Report
After revising the new version of the manuscript “Molecular mechanisms of phenylpropane-synthesis-related genes regulating the shoot blight resistance of Bambusa pervariabilis × Dendrocalamopsis grandis”, this reviewer has seen that most of the suggestions and comments have been addressed by the authors. Only some comments are provided regarding their revision:
- The authors have shortened the Abstract as required, since it was too long according to the Author’s guidelines. Although it is shorter now, it is still a little long, with more than 200 words. The authors could decide, in accordance with editor, if they could keep it this way or they should shorten it once more.
- Line 25, please correct the word ‘verieties’ for ‘varieties’.
- The authors have included some more information in the Introduction, as suggested. However, although that information is good, it is very general and related only with the importance of breeding plant resistance. It would be nice if the authors could include some information regarding the genetic resources of plant resistance available in that plant species against that disease. It means, if there are few or a lot of resistant varieties available to be used.
- In Materials and methods: regarding the comments done by this reviewer (line 470 in the previous version: “could you specify the information about de standard curve that you name ‘dissolution curve’? It is advisable that the authors follow The MIQE Guidelines (Bustin et al., 2009) for qPCR experiments”), it is not clear the revision carried out by the authors. They said the following: “they want to show that the Melt Peak graph is output after the qPCR reaction. Due to translation problems, ambiguity is caused. We decided to delete this ambiguous sentence from the revised manuscript”. However, it is not clear which sentence they have deleted. Besides, they do not explain anything about the dilution curve for the qPCR. I would be grateful if they could explain clearly this point.
Reviewer 3 Report
The authors incorporated most of the review’s suggestions to the manuscript. The manuscript quality greatly improved and it is suitable for publication in the International Journal of Molecular Sciences.
Round 3
Reviewer 2 Report
After checking the revised version of the manuscript “Molecular mechanisms of phenylpropane-synthesis-related genes regulating the shoot blight resistance of Bambusa pervariabilis × Dendrocalamopsis grandis”, this reviewer has seen that the suggestions and comments have been addressed by the authors.